# Air Quality, Pollution and Sustainability Trends in South Asia: A Population-Based Study

**DOI:** 10.3390/ijerph19127534

**Published:** 2022-06-20

**Authors:** Saima Abdul Jabbar, Laila Tul Qadar, Sulaman Ghafoor, Lubna Rasheed, Zouina Sarfraz, Azza Sarfraz, Muzna Sarfraz, Miguel Felix, Ivan Cherrez-Ojeda

**Affiliations:** 1Research, Nishtar Medical University, Multan 60000, Pakistan; saimarao95@gmail.com (S.A.J.); drsalmanlashari@gmail.com (S.G.); 2Research, Quaid-e-Azam Medical College, Bahawalpur 06318, Pakistan; maliklaila23@gmail.com; 3Research, University Medical and Dental College Faisalabad, Faisalabad 38800, Pakistan; mubashargreewal@gmail.com; 4Research and Publications, Fatima Jinnah Medical University, Queen’s Road, Lahore 54000, Pakistan; 5Pediatrics and Child Health, The Aga Khan University, Karachi 74800, Pakistan; azza.sarfraz@aku.edu; 6Research, King Edward Medical University, Lahore 54000, Pakistan; sarfrazmuzna@gmail.com; 7Allergy and Pulmonology, Universidad Espíritu Santo, Samborondón 0901-952, Ecuador; miguel.felixromero@gmail.com; 8Respiralab Research Center, Guayaquil 0901-952, Ecuador

**Keywords:** air quality, air pollution, sustainability, South Asia, population trends

## Abstract

Introduction: Worsening air quality and pollution lead to numerous environmental health and sustainability issues in the South Asia region. This study analyzes India, Nepal, Bangladesh, Pakistan, Sri Lanka, and Nepal for air quality data trends and sustainability indicators. Methodology: By using a population-based study design, six South Asian countries were analyzed using a step-wise approach. Data were obtained from government websites and publicly available repositories for region dynamics and key variables. Results: Between 1990 and 2020, air quality data indicated the highest rise in CO_2_ emissions in India (578.5 to 2441.8 million tons) (MT), Bangladesh, Nepal, and Pakistan. Greenhouse gas emissions, from 1990 to 2018, nearly tripled in India (1990.4 to 3346.6 MT of CO_2_-equivalents), Nepal (20.6 to 54.6 MT of CO_2_-equivalents), and Pakistan, and doubled in Bangladesh. Methane emissions rose the highest in Pakistan (70.4 to 151 MT of CO_2_-equivalents), followed by Nepal (17 to 31 MT of CO_2_-equivalents) and India (524.8 to 669.3 MT of CO_2_-equivalents). Nitrous oxide nearly doubled in Bangladesh (16.5 to 29.3 MT of CO_2_-equivalents), India (141.6 to 256.9 MT of CO_2_-equivalents), Nepal (17 to 31 MT of CO_2_-equivalents), and more than doubled in Pakistan (27 to 61 MT of CO_2_-equivalents). On noting particulate matter 2,5 annual exposure, India saw the highest rise from 81.3 µg/m^3^ (in 1990) to 90.9 µg/m^3^ (2017), whereas trends were steady in Pakistan (60.34 to 58.3 µg/m^3^). The highest rise was noted in Nepal (87.6 to 99.7 µg/m^3^) until 2017. During the coronavirus disease 19 pandemic, the pre-and post-pandemic changes between 2018 and 2021 indicated the highest PM_2.5_ concentration in Bangladesh (76.9 µg/m^3^), followed by Pakistan (66.8 µg/m^3^), India (58.1 µg/m^3^), Nepal (46 µg/m^3^) and Sri Lanka (17.4 µg/m^3^). Overall, South Asian countries contribute to the worst air quality and sustainability trends regions worldwide. Conclusions: Air pollution is prevalent across a majority of South Asia countries. Owing to unsustainable industrial practices, pollution trends have risen to hazardous levels. Economic, environmental, and human health impacts have manifested and require urgent, concerted efforts by governing bodies in the region.

## 1. Introduction

Worsening air quality and pollution pose numerous sustainability and environmental health challenges in South Asia—home to countries vulnerable to climate change [1]. India, Nepal, Bangladesh, and Pakistan are four of the most polluted countries worldwide. This study incorporates Bangladesh, Bhutan, India, Pakistan, Nepal, and Sri Lanka as epicenters of ambient air pollution in the South Asian region [1]. A consequence of air pollution—the “brown cloud” in cities such as New Delhi and Lahore is typically caused by carbon aerosols; this phenomenon has been captured via satellite images of atmospheric haze [2]. Certain cultural practices such as using cooking stoves in rural and semi-urban households further contribute to the highest concentrations of black carbon emissions [3]; industrial emissions, cars, and trucks also contribute to aggravated carbon emissions [2,4]. As per the World Air Quality Report in 2020, 37 out of 40 of the top-most polluted cities in the world are in South Asia [1]. Since 2010, around 700 million people, constituting half of the South Asian population, have been impacted by at least one climate-related disaster [1]. The diminished air quality is also expected to directly affect over 800 million people in the region by 2050 and burden the economy [1]. Air pollution is the second-highest risk factor for adverse health outcomes in the region [5].

Moreover, air pollution is the third-highest risk indicator for premature death in South Asia compared to the night-ranked cause in Western Europe [5]. Air pollution also contributes to 11% of deaths, leading to 40 million disability-adjusted life years in South Asia [5]. However, air pollution is not a localized notion and is transported across countries and borders, impacting sustainability, water resources, and human lives [2]. Concerted regional actions are required to monitor air quality in South Asia while implementing evidence-based interventions [6]. Countries have initiated promising initiatives in recent years, but health-centric strategies are lacking, as documented by adverse health and environmental outcomes [4]. A study identifies the effects of ambient air pollution on pregnancy losses in South Asia, with an estimated 349,681 pregnancy losses reported each year across Bangladesh, India, and Pakistan associated with air pollution [7]. These findings are pertinent for the South Asian region—one of the most polluted regions worldwide—and to improve both maternal and maternal health in low-income countries [7]. Current literature also links air pollution to increased miscarriages, low birth weight among infants, and premature births due to pollution-related adverse events in the mother [8]. Moreover, one study also identifies that air pollution may be linked to breached mother’s placentas and can breach the fetus as well [9]. Previously published reports on maternal and public health outcomes are pertinent as these may indirectly have mental, physical, and economic effects on the population; these include but are not limited to postnatal depression, infant mortality, increased costs related to pregnancy, and loss of labor [7,9].

Our current understanding, based on scholarly reports and organizational reports, is that air pollution is a major environmental health risk. The World Health Organization (WHO) reports that 4.2 million die each year, across the world, due to heart disease, stroke, lung cancer, and acute/chronic respiratory disease linked to ambient air pollution [10]. This can be further supported by WHO data suggesting that 99% of the world’s population is living in localities where the WHO air quality guidelines were unmet in 2019 [10]. Furthermore, 91% of premature deaths occur in low- and middle-income countries due to air pollution-induced environmental effects in South-East Asia [10]. Another 2013 assessment led by WHO’s International Agency for Research on Cancer found that air pollution particles are carcinogenic to humans, with particulate matter components most likely associated with the increase in cancer incidence, particularly lung cancer [11]. At this juncture, it is clear that most sources of air pollution are beyond individual control and demand concerted efforts by regional, national, and local policy-makers working in sectors such as health, waste management, energy, transport, urban planning, and agriculture [12].

This study will present the trends of air quality data and sustainability indicators in South Asian countries and provide recommendations for collective national and international action plans with an intersectoral approach.

## 2. Methods

Data were monitored between 1990 to 2021 to understand the trends among South Asia countries. The methodology used in this study has been previously applied in literature [13,14]. The population-based study obtained data from national and international data banks and government websites (Data on CO_2_ and Greenhouse Gas Emissions by Our World in Data—https://www.worldbank.org/en/home (accessed on 3 May 2022); World Development Indicators—http://wdi.worldbank.org/table/WV.3# (accessed on 3 May 2022); CO_2_ and Greenhouse Gas Emissions dataset—https://github.com/owid/co2-data (accessed on 3 May 2022)). The CO_2_ and Greenhouse Gas Emissions dataset is a collection of metrics maintained by ‘Our World in Data.’ The Global Carbon Project releases CO_2_ emissions annual data. Greenhouse gas emissions (nitrous oxide and methane) data were sourced from the CAIT Climate Data Explorer. Energy data were sourced from the BP Statistical Review of World Energy, which only provides information on primary energy consumption. The World Bank Development Indicators and the World Bank total population figure data were used for any additional information. Other variables were collated from a variety of sources such as the United Nations, Gapminder, Maddison Project Database, and the World Bank. The data are attached under Appendix A. Other sources were utilized, including peer-reviewed articles on air trends within South Asia. Search terms were used with Boolean operators and included “South Asia”, “Air”, “Pollution”, “Sustainability”, and “Health”. Studies/reports/data repositories that addressed air quality, pollution, or health were included.

A classification list of the terminology used throughout the analysis is presented as follows:(1)*Carbon dioxide* (*CO_2_*): Annual production-based emissions of carbon dioxide (CO_2_), measured in million tons.(2)*Total greenhouse gasses*: Total greenhouse gas emissions, including land-use change and forestry, measured in million tons of carbon dioxide-equivalents.(3)*Methane*: Total methane emissions including land-use change and forestry, measured in million tons of carbon dioxide-equivalents.(4)*Nitrous oxide* (*NO_2_*): Total nitrous oxide emissions, including land-use change and forestry, measured in million tons of carbon dioxide-equivalents.(5)*Population-weighted exposure to ambient PM_2.5_*: The average level of exposure the country’s residents have to concentrations of suspended particles measured less than 2.5 µg in the aerodynamic diameter; these particles are capable of penetrating the respiratory tract and lead to severe health damage. The exposure is calculated by weighting the mean annual concentration in populations spanning rural and urban areas.(6)*Nationally protected terrestrial and marine areas*: These are terrestrial protected areas that are partially or totally protected, at least 1000 hectares as designated by the national authorities. Marine protected areas include intertidal or subtidal terrain that overlies water and has historical and cultural features.(7)*Renewable energy consumption*: The share of renewable energy in total final energy consumption, reported as the percentage of total final energy consumption.(8)*Access to electricity*: Percentage of population with access to electricity.(9)*People using safely managed sanitation services*: Percentage of the population using improved sanitation facilities not shared with other households where excreta are safely disposed.(10)*People using safely managed drinking water services*: Percentage of the population utilizing drinking water from a source that is accessible, available as required, and free from fecal and priority chemical contamination.

The raw data of all 6 countries (Bangladesh, India, Pakistan, Nepal, Bhutan, and Sri Lanka) were collected for air quality, pollution, and sustainability trends and processed country-wise. The earliest yearly report was obtained for 1990 and wherever available until 2021 for the latest year. Herein, air quality was assessed by collecting CO_2_, greenhouse gas, methane, and NO_2_ data. Air pollution was reported with PM_2.5_ data in raw and with population-weighted exposure. Sustainability trends were assessed for the following: nationally protected terrestrial and marine areas, renewable energy consumption, access to electricity, and population using safely managed sanitation/drinking water services.

The primary outcome of this study was to provide estimates of trends and shifts since 1990 for the myriad of air quality and pollution factors. The secondary outcome was to explore sustainability trends in recent years to inform country-wise and regional responses in the public health domain. This study was exempt from the Institutional Review Board (IRB) committee approval due to the secondary nature of the analysis and sourcing of publicly available data. The step-by-step approach is attached in Figure 1.

## 3. Results

### 3.1. Air Quality Data

Since 1990, the CO_2_ emissions in Bangladesh have increased more than six-fold from 14.08 to 92.8 million tons in 2020. India emitted 578.5 million tons of CO_2_ in 1990; the 2020 yearly report finds emissions valuing 2441.8 million tons, which increased four-fold over 30 years. While Nepal was producing 0.72 million tons of CO_2_ in 1990, the emissions rose to 16.97 million tons in 2020, which is an estimated 24 times rise. Pakistan emitted 67.8 million tons of CO_2_ in 1990, but 2020 trends found gross values of 234.8 million tons, which is a 3.5-fold increase. Sri Lanka and Bhutan had lower CO_2_ emissions than the other countries, with 21.1 million tons (5.5-fold rise since 1990) and 1.93 million tons (15-fold rise since 1990), respectively, in 2020 (Figure 2). The highest cumulative total of CO_2_ emissions in 2020 only was documented in India, followed by Pakistan and Bangladesh. However, the largest rise (in 2020 compared to 1990) was noted in firstly Nepal (23.6-fold), secondly, Bhutan (15-fold), and lastly, Bangladesh (6.6-fold).

Greenhouse gas emissions doubled in Bangladesh from 1990 (115 million tons of CO_2_-equivalents) to 2018 (220.75 million tons of CO_2_-equivalents), a nearly two-fold rise. India produced 1009.4 million tones of greenhouse gas emissions in 1990, and the output nearly tripled by 2018 (3346.6 million tons of CO_2_-equivalents). Nepal’s greenhouse gas emissions increased from 20.6 million tons of CO_2_-equivalents in 1990 to 54.6 million tons of CO_2_-equivalents in 2018, a nearly 2.7-fold rise. Pakistan reported increases in greenhouse gas emissions from 166.5 million tons of CO_2_-equivalents in 1990 to 438.2 million tons of CO_2_-equivalents in 2018, a 2.6-fold increase. The lowest increase was noted in Bhutan, which reported negative 5.46 million tons of CO_2_-equivalents of emissions in 1990 and 1.16 million tons of CO_2_-equivalents in 2018. Sri Lanka increased from 29.2 to 37.15 million tons of CO_2_-equivalents from 1990 to 2018, a nearly 1.3-fold rise (Figure 2). The highest cumulative total of greenhouse gas emissions in 2020 only was reported in India, followed by Pakistan and Bangladesh. However, the largest rise (in 2018 compared to 1990) was noted in firstly India (3.3-fold), secondly Pakistan (2.6-fold), and thirdly Bangladesh (1.9-fold).

Total methane emissions in Bangladesh spanning 1990 to 2018 increased from 64.43 to 83.81 (1.3-fold rise) million tons of CO_2_-equivalents. In India, methane emissions increased from 524.8 to 669.3 (1.3-fold rise) million tons of CO_2_-equivalents. Nepal’s methane emissions increased from 17 to 31 (1.8-fold rise) million tons of CO_2_-equivalents from 1990 to 2018. Pakistan reported changes from 70.4 to 151 (2.2-fold rise) million tons of CO_2_-equivalents from 1990 to 2018. Bhutan was the only South Asian country with no changes in emissions (0.85 to 0.86 million tons of CO_2_-equivalents). In contrast, Sri Lanka was the only country that had reduced methane emissions (18.4 to 10 million tons of CO_2_-equivalents) (Figure 2). The highest cumulative total of methane emissions in 2020 only was reported in India, followed by Pakistan and Bangladesh. However, the largest rise (in 2018 compared to 1990) was noted firstly in Pakistan (2.2-fold), secondly in Nepal (1.8-fold), and lastly in Bangladesh/India equally (1.3-fold rise).

Nitrous Oxide emissions increased from 16.5 to 29.3 (1.8-fold rise) million tons of CO_2_-equivalents in Bangladesh from 1990 to 2018. India’s emissions increased from 141.6 to 256.9 (1.8-fold rise) million tons of CO_2_-equivalents between 1990 and 2018. Nepal’s nitrous oxide emissions also nearly doubled from 17 to 31 (1.8-fold rise) million tons of CO_2_-equivalents from 1990 to 2019. Pakistan’s emissions more than doubled from 27 to 61 (2.3-fold rise) million tons of CO_2_-equivalents. Bhutan and Sri Lanka were the two countries with no changes in emissions between 1990 and 2018 (0.18 to 0.19 and 2.25 to 2.24 million tons of CO_2_-equivalents, respectively) (Figure 2). The highest cumulative total of nitrous oxide emissions in 2020 only was reported for India, followed by Pakistan and Nepal. However, the largest rise (in 2018 compared to 1990) was noted firstly in Pakistan (2.3-fold), followed by a similar rise in Bangladesh, India, and Nepal (1.8-fold rise each).

### 3.2. PM_2.5_ Air Pollution

The mean annual exposure to PM_2.5_ was steady in Pakistan (60.34 µg/m^3^ in 1990 and 58.3 µg/m^3^ in 2017). However, the mean annual exposure in India rose from 81.3 in 1990 µg/m^3^ to 90.9 µg/m^3^ in 2017. Nepal’s exposure rose to PM_2.5_ rose from 87.6 µg/m^3^ in 1990 to 99.7 µg/m^3^ in 2017. Sri Lanka (29.8–11.1 µg/m^3^) and Bhutan (40.2–38 µg/m^3^) had no increases in annual exposures (Figure 3).

### 3.3. Annual Change in PM 2.5 Concentrations Pre- and Post-COVID-19

The annual shifts in PM_2.5_ concentration between 2018 and 2021 are depicted in Figure 4. Bangladesh reported a reduction of 20 µg/m^3^ between 2018 to 2020. Notably, Bangladesh had the largest PM_2.5_ concentration between 2018–2021, whereas Pakistan showed a reduction of 15.3 µg/m^3^, followed by a rise of 7.8 µg/m^3^ in 2021; Pakistan had the second-worst PM_2.5_ concentration in South Asia. India was the third-worst country in particulate matter concentrations, with reductions of 20.6 µg/m^3^ between 2018 to 2020; the levels rose by 6.2 µg/m^3^. Nepal and Sri Lanka had overall low PM_2.5_ concentrations both before and during the COVID-19 pandemic (Figure 4).

### 3.4. Sustainability Trends

In 2020, the percentage of people using safely managed to drink water services was lowest in Nepal (17.6%), followed by Pakistan (35.8%) and Bhutan (36.6%). Data could not be obtained for India and Sri Lanka, whereas in high-income countries, 97.7% of people could safely manage drinking water services. In 2020, the percentage of the population safely managing the use of sanitation services was the lowest in Bangladesh (38.7%), followed by India (45.9%), Nepal (48.6%), and Bhutan (65.2%). Compared to South Asia, 87% of high-income country residents could manage sanitation services in 2020. In 2017, on noting access to electricity, the least population percentage was noted in Pakistan (70.8%), followed by Bangladesh (88%), India (92.5%), Sri Lanka (97.5%), and Bhutan (97.7%). In comparison, virtually 100% of high-income countries had access to electricity in 2017. In 2015, on noting renewable energy consumption, the percentage of total final energy consumption was the highest in Bhutan (86.7%), followed by Nepal (85%), Sri Lanka (52.9%), Pakistan (45.9%), India (34.4%) and Bangladesh (34.2%) in 2015. The percentage was lower in high-income countries (11.4%). Finally, in 2018, the percentage of total area protected by terrestrial and marine area was the highest in Bhutan (48%), followed by Nepal (23.6%), Pakistan (9.8%), Bangladesh (4.9%), India (3.5%), and Sri Lanka (3.4%). High-income countries reported 19.1% nationally protected terrestrial and marine areas (Figure 5).

Additionally, in 2016, the ambient air pollution of PM_2.5_, expressed as micrograms per cubic meter, was highest in Nepal (98.1 µg/m^3^), followed by India (89.7 µg/m^3^), Bangladesh (60.1 µg/m^3^), Pakistan (58.6 µg/m^3^), Bhutan (37.2 µg/m^3^), and Sri Lanka (14 µg/m^3^). The mean annual exposure in high-income countries was low at 14.5 µg/m^3^.

## 4. Discussion

This study is the first original paper to assess air quality, pollution, and sustainability in the South Asian region using population-based analytical techniques. The main findings of this paper are as follows. First, on assessing for air quality changes between 1990 and 2020, CO_2_ emissions increased by 23.6-fold in Nepal, 15-fold in Bhutan, and 6.6-fold in Bangladesh. However, India had the highest burden of CO_2_ emissions in 2020, which was followed by Pakistan and Bangladesh. Greenhouse gas emissions saw the largest increase in India (3.3-fold), with a 2.6-fold increase in Pakistan and a 1.9-fold rise in Bangladesh. Overall, India had the highest cumulative total of greenhouse gas emissions. Moreover, methane emissions rose the largest in Pakistan, by 2.2-fold, followed by Nepal (1.8-fold) and India/Bangladesh (1.3-fold rise). NO_2_ emissions were increased by 2.3-fold in Pakistan, with 1.8-fold rises in India, Bangladesh, and Nepal each.

Second, when measuring the ambient air pollution of PM_2.5_, in 2016, the highest presence was in Nepal, followed by India and Bangladesh. Air pollution shifts were also reported as population-weighted exposure to ambient PM_2.5_ pollution. Overall, rises in mean annual PM_2.5_ exposures were reported in India and Nepal only between 1990 and 2017; the other four countries had stable PM_2.5_ concentrations until 2017. The annual PM_2.5_ concentration between 2019 and 2020 was also ascertained to note pre- and post-COVID-19 pandemic fluctuations; Pakistan had the highest reduction of annual PM_2.5_ (6.8 µg/m^3^), whereas Bangladesh and India had comparable reductions (6.2 µg/m^3^).

Third, sustainability trends were assessed for various actions, including safely managing drinking water—to which only 17.6% of people in Nepal had access, followed by 35.8% in Pakistan and 36.6% in Bhutan. Sanitation service access was the lowest in Bangladesh (38.7%), which was followed by 45.9% having access in India and 48.6% with access in Nepal. Further, electricity access was the lowest in Pakistan (70.8%), followed by 88% of the Bangladeshi population and 92.5% in India.

On reviewing current literature for published reports assessing air pollution and trends in South Asia, there is a dearth of data. For instance, Krishna and colleagues provide their perspectives on air pollution being a central risk factor for adverse health in South Asia [2]. In their report, the interconnectedness of the South Asian airshed is highlighted, which further requires regional cooperation by international governments. Krishna et al. also found that a systematic collection of air quality data is essential to improve air quality management [2]. In line with the recommendations the authors made in 2017 [2], this study systematically collates air pollution trends by analyzing multiple indicators in the region, thereby filling pertinent gaps in current literature. A scoping study for South Asia’s air pollution prepared by the Energy and Resource Institute (TERI, India) assesses the current state of evidence of air pollution in all South Asian countries and identifies gaps and priority areas at the regional and country-level [21]. As stated in the scoping study, air monitoring in the South Asian region has largely evolved over the years as strategies have been placed to mitigate the adverse events of pollution on human health [21]. Certain areas of improvement in gaps and recommendations include (i) population awareness-building activities to reduce indoor air pollution, (ii) data auditing, (iii) analytical quality control, (iv) public participation, and (v) promotion and implementation of public awareness programs [21].

A modeling study assessing the stipulated impacts of air quality changes by 2050 reports the following; the progression of air pollution (PM_2.5_ and ozone) requires evaluation of the different social/economic pathways in the South Asia region to assess the implemented/planned Government policies on air quality and thereby present estimates of future air quality in South Asia [12]. These actions will improve emission mitigation strategies. Our study additionally impinges on sustainability trends to provide more information on the country-wise fulfillment of sustainable development goals (i.e., access to water, sanitation, electricity, etc.) [22]. A brief report by Mishra et al. in 2021 reviewed changes in air pollution in South Asia during the COVID-19 lockdown situation [23]. The study determined that major improvements were witnessed in air quality in populated cities such as Delhi, Kathmandu, Dhaka, and Colombo, in addition to Islamabad [23]. The report further instills the notion that with strict lockdown impositions and reduced use of large-scale transport and industrial emissions, harmful pollutants are reduced [23]. As with the report [23], our study provides recommendations on proposing an inter-country agency that can provide optimum data control and balance both energy and resource consumption as we progress across multiple phases of COVID-19 [24]; the key target area includes sustainability indicators, as highlighted in our study. Hasnat et al. (2018) perform a South Asian analysis to discuss the major environmental issues faced in the region, with a special focus on Bangladesh. The key issues identified by the authors include climate change, ecosystem changes, population pressure, water resources, energy resources, degradation of marine and river resources, and pollution of land resources; in our study, we identify access to drinking and sanitation water and access to electricity as central to promoting a sustainable South Asian region [25].

On study country-level outputs, Bangladesh served as the most polluted country on average, followed by Pakistan and India [26]. Notably, China—an East Asian authoritarian state—has made excellent strides in improving air quality with policy and legal regulations [26]. In the 2017–2018 Chinese report, the average PM_2.5_ concentrations fell by 12%, representing Beijing’s trends—the country’s capital, which has reduced by over 40% since 2013 [26]. A 2018 report by the United Nations Environment Program also finds that toxic air kills seven million worldwide annually—the largest contributions occur in the Asian-Pacific region [27]. A study published in Lancet Planetary Health also finds that 1.24 million died from air pollution in 2017 [28,29]. Current evidence suggests that the highest recorded levels of air pollution are in the Asian Pacific countries. At present, 2.3 billion people in the region are exposed to hazardous air, with India presenting with annual mean PM_2.5_ concentrations of 40 µg/m^3^ [30], whereas Bangladesh and Pakistan belong to the WHO interim target three groups with averages of 15 µg/m^3^ yearly mean PM_2.5_ concentrations [30]. The most deadly air pollutants, while subjective, are ground-level ozone and fine particulate matter (PM_2.5_) [31]. Heavily industrialized areas with high population densities have the most increased air pollution. In 2015 suggested that 33% of the global deaths from outdoor air pollution occurred in South Asia [30]. In addition to impacting well-being and human health, air pollution compromises the quality of agricultural produce and food security in this region, where 60% of the global population of undernourished groups resides [30].

As the world recovers from the coronavirus disease 19 (COVID-19) pandemic, public health and promotion agencies must note improved air quality in South Asia [32]. A geographical representation of average PM_2.5_ levels in the year 2020 is presented in Figure 6. The overall reduction of AQI after the lockdown period in 2020, compared to 2019, in South Asia is reported for Delhi (<41%), Dhaka (<16%), Kathmandu (<32%), and Colombo (<33%) [23]. Up to 50% reductions have been observed in NO_2_ levels in South Asia during the lockdown [23]. The reduced pollutant levels can be attributed to altering fossil fuel emissions. The lockdown period has paved a new pathway for sustainable development in South Asia. The concentrations of gaseous pollutants and particulate matter decreased drastically in the initial lockdown phase due to the stoppage of outdoor activities. These actions led to a reduction of 50% in the air quality index (AQI) in South Asian cities. Reports find that Delhi (41%), Kathmandu (32%), Dhaka (16%), Colombo (33%), and Islamabad showed gross reductions in CO, SO_2_, and NO_2_ [23]. The lockdowns provided ample opportunities for authorities to reassess large-scale air polluting transport and industrialized localities, provided the large emissions of pollutants [33]. Trends also reveal a need to control emissions by adjusting to a new normal by switching to cleaner fuel technology options. An inter-state agency would be helpful in addition to maintaining a balance between energy and resource consumption to improve air quality and sustainability throughout the region [33].

## 5. Health Impact of Air Pollution

Exposure to high rates of air pollution has resulted in air pollution being the second most important risk factor for non-communicable diseases globally after tobacco smoking [34]; in South Asia, it is now a leading risk factor of non-communicable disease [35]. It is responsible for 58 million disability-adjusted life years (DALYs) due to the increasing frequencies of cancer, stroke, and chronic and acute respiratory diseases, including asthma, associated with air pollution [35]. The International Agency for Research on Cancer has classified air pollution as carcinogenic [36]. Particulate matter in polluted air is of small variable diameters that enter the respiratory system through inhalation [37]. When large quantities of air pollutants enter the human body, they lead to direct poisoning and chronic intoxication [4]. Smaller particulate matter is potentially associated with more severe morbidity as it may be able to penetrate the airways easily to reach the bloodstream [38]. Short-term exposure to air pollutants causes symptoms similar to chronic obstructive pulmonary disease (COPD), such as cough, shortness of breath, wheezing, respiratory disease, and high hospitalization rates [39,40]. A physical manifestation of acute pollutant injury is voice alteration due to contamination in the trachea [41]. Other short-term manifestations include hypertension, stroke, myocardial infarction, and heart insufficiency in the cardiovascular system with high exposure to traffic emissions [42,43].

The long-term effects are more frequently manifested among individuals with a predisposing disease state [44]. Long-term exposure causes adverse health effects, particularly in vulnerable age groups such as the elderly and children, on respiratory, cardiovascular, and neurological systems [38,45]. Long-term exposure to nitrous oxide has been linked with ventricle hypertrophy accompanied by underlying changes in blood cells [46,47]. Recently, neurodegenerative diseases such as Alzheimer’s and Parkinson’s have been connected with air pollution in a dose-dependent manner [48]. Underlying mechanisms implicated are at the neuronal level associated with oxidative stress, protein accumulation, inflammation, and mitochondrial impairment [49]. More recently, a Swedish cohort found a prominent association between air pollution categorized by PM10 and nitrogen dioxide exposure and diabetes mellitus [50]. In children, air pollution exposure is associated with compromised fetal growth and low birth weight, poor respiratory system development, respiratory infections, and autism [51,52,53,54]. Socioeconomic vulnerability is another factor that predisposes individuals to adverse health impacts due to housing, diet, genetics, and education [55,56,57,58]. Emerging evidence suggests that air pollutants dysregulate the immune system at the innate immunity level (e.g., macrophage costimulatory molecule CD80 and CD86 upregulation) [4,59]. While the major health effects of air pollution are via inhalation, skin penetration of pollutants (e.g., oxides and photochemical smoke) may contribute to psoriasis, urticaria, eczema, atopic dermatitis, and skin cancer [60,61]. The eye is vulnerable to damage from air pollutants resulting in retinopathy in severe cases [62,63].

## 6. Ongoing Initiatives

Vehicles are the primary cause of air pollution in urban cities across South Asian countries [23]. Bangladesh was the first South Asian country to stop adding lead to gasoline, performed historically to boost octane inexpensively [64]. Other South Asian countries similarly stopped using lead in gasoline, including India, Nepal, Pakistan, and Sri Lanka [65]. Bangladesh is focusing on raising awareness, transportation planning, fuel reformation, and regulatory measures [6]. By the early 2000s, Bangladesh started regulating the use of two-stroke engine three-wheelers and diesel vehicles which were the leading cause of smoke and hydrocarbon emissions [66]. At present, no new two-stroke engine three-wheelers have been registered in the past few years, leading to decreased concentration of fine PM and black carbon concentrations [66]. Bangladesh is focusing on expanding CNG use as fuel for vehicles with help from the Asian Development Bank, a feasible option as the country is home to a high quantity of natural gas reserves [67]. Air quality management is being regulated to tackle air pollution [68]. India has imposed various policies to control air pollution specific to industries with a high level of pollution, traffic flow, and regulation of community-level pollution [6]. India set up its first systematic air quality monitoring in 1967 across 10 cities by the National Environmental Engineering Research Institute (NEERI) [65]. The Indian government is prioritizing its largest metropolitan cities—Chennai, Delhi, Kolkata, and Mumbai [69]. A notable initiative was taken by the Indian Supreme Court to address the worsening air pollution in Delhi [70]. Interventions to relocate industries with high pollution rates outside the city were conducted [71]. Public transport was mandated to be run on CNG by March 2001 [72]. Other initiatives currently in motion are enforcing regulatory standards and rationalization of fuel taxes [73].

Nepal is focusing on the Kathmandu valley as air pollution remains high [52,74]. Due to thermal inversion, the weather conditions are harsh in Kathmandu and other valleys, especially in winters [75]. Nepal’s challenges in controlling air pollution are similar to those of Bangladesh, which has limited capacity to manage air pollution. Certain industries, such as cement and brick-making, are the highest contributors to pollution in Kathmandu valley [76,77]. The government has also issued emission standards for in-use vehicles since 2000 [78]. Overall, capacity building for monitoring and regulation is required. Pakistan has high natural gas reserves, similar to Bangladesh, and the Hydrocarbon Development Institute of Pakistan (HDIP) is promoting the use of CNG as a cleaner alternative to transportation fuel [79]. However, diesel remains a frequent option for vehicle fuel as CNG is less economical [80,81]. Pakistan piloted an emissions inspection system for vehicles with help from the UNDP/GEF fundings for the Road Transport Sector Project in 2005 [82]. Similar to Bangladesh and Nepal, air quality monitoring in Pakistan is not optimal as the implementation challenges are financial but require further attention as the air quality situation is critical [6,83]. Sri Lanka has taken a more proactive approach by fining vehicles with high emission loads, encouraging low emission vehicles, fuel reformation, and transportation planning [6]. The Clean Air 2000 Action Plan was implemented, which was declared unsuccessful due to financial limitations [84]. Recently, Air Resource Management Center (AirMac) was activated to address challenges to vehicle emission regulation [6].

## 7. Recommendations

While existing policies are working toward reducing pollution, they do not address population-weighted exposure to harmful air matters that could grow by more than 50% by 2030, based on the anticipated economic group in the period [30]. Provided the urbanization and growing economy, air pollution may expose around four billion people to health-damaging pollution levels [10,85]. By incorporating technological solutions and clear air measures, over 650,000 may breathe clean air by 2030 [30]. A key challenge common within the South Asian region is air quality management in urban settings [2]. Setting up air quality monitoring sites requires huge financial investments that may meet challenges as all South Asian countries are low- and middle-income [86]. Once monitoring sites are set up across highly polluted cities, there needs to be a designated team that oversees data analysis and regulation [87]. While automated instruments require less human overseeing, it is a costlier undertaking; recent low-cost sensors are being explored but have several challenges such as cross-reactivity with ambient pollution, traffic, weather, and declining accuracy with time [88].

The important question is to streamline which industry is a significant contributor to the rising air pollutant burden [89]. The prime focus of the governments has remained on vehicle exhaust emissions across South Asia [90]. India has had the longest-standing vehicle inspection system in South Asia, yet there has been sub-optimal improvement in the air quality [91]. A channel between national and provincial governments can help enhance the country’s singular efforts by different states [92]. All South Asia countries have already authorized task forces with a prime focus on vehicle emission regulations [93]. Other agenda items are traffic management, industrial pollution, and community-level pollutants such as wood/fuel burning and garbage burning [94,95]. Ideally, mapping technologies may be able to determine epicenters within heavily-polluted urban cities in South Asia [96]. Other sectors must also be educated and sensitized to ensure that emission levels are acceptable environmentally [97,98]. Regular inspections must be made compulsory across industrial sites with high emission rates and vehicle emissions [99]. An annual vehicle inspection of all in-use vehicles across South Asia can help control air pollution [100]. A combined technical assistance task force may be established to learn from one another’s experience thus far and further enhance monitoring standards, implementation, and expansion of current capacity.

## 8. Conclusions

Air pollution is highly prevalent across all South Asian countries. Pollution levels have risen in the last few decades due to economic growth and industrialization in these countries. However, vehicle and stack emissions from unsustainable industrial practices have caused air pollution to rise to hazardous levels. Human health, economic, and environmental impacts of the alarmingly high air pollution levels require urgent and concerted efforts from South Asia.

## Figures and Tables

**Figure 1 ijerph-19-07534-f001:**
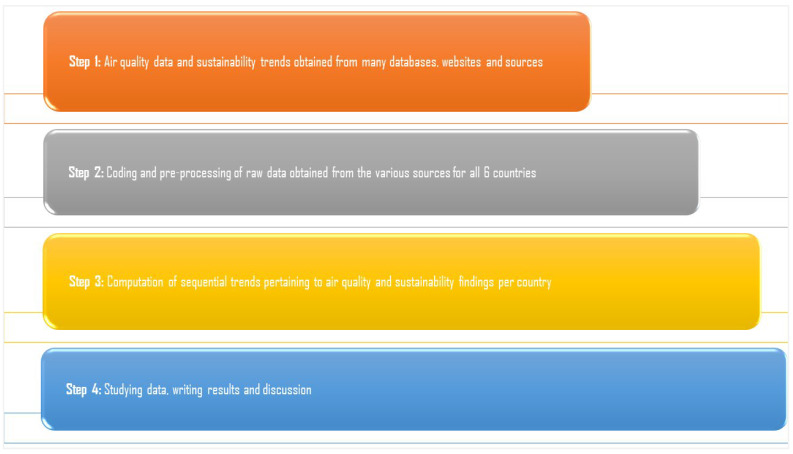
Step-wise breakdown of the employed methodology of epidemiological analysis.

**Figure 2 ijerph-19-07534-f002:**
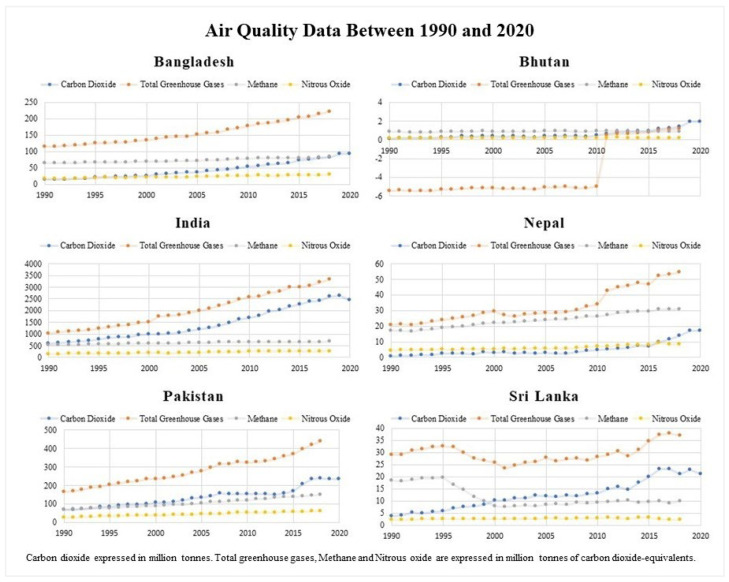
Air Quality Data Between 1990 and 2020 [15]. CO_2_ is expressed in million tons. Total greenhouse gases, methane, and NO_2_ are all expressed in million tons of carbon dioxide-equivalents.

**Figure 3 ijerph-19-07534-f003:**
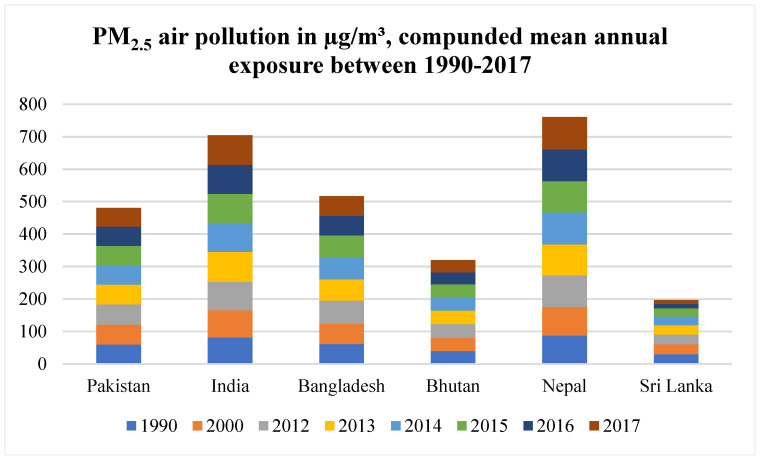
Population-weighted exposure to ambient PM_2.5_ pollution expressed in µg/m^3^ [16].

**Figure 4 ijerph-19-07534-f004:**
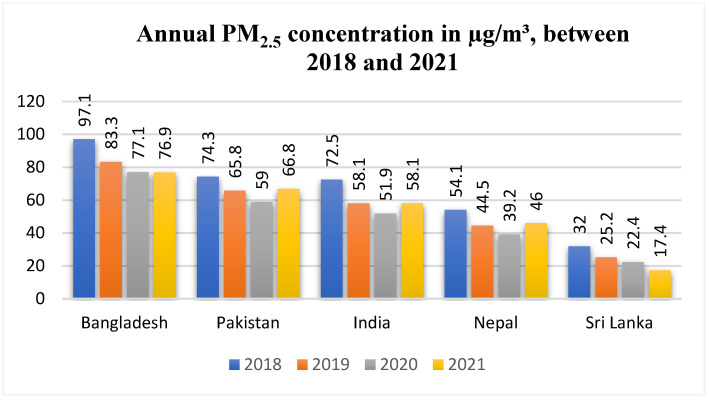
Annual means of PM_2.5_ concentrations, depicted from 2018 to 2021 in µg/m^3^ [17]. Data were unavailable for Bhutan.

**Figure 5 ijerph-19-07534-f005:**
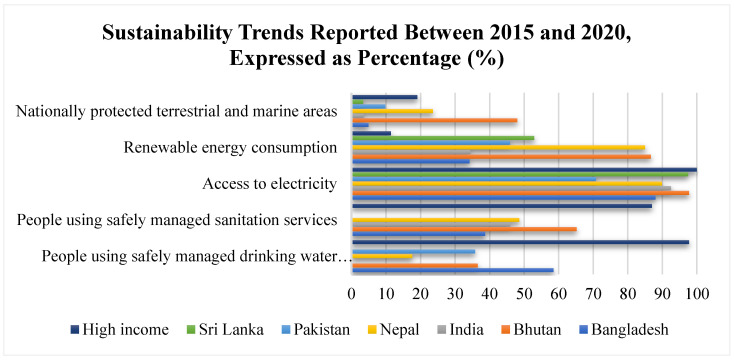
Sustainability Trends reported in the years between 2015 and 2020 reported as a percentage [18,19,20].

**Figure 6 ijerph-19-07534-f006:**
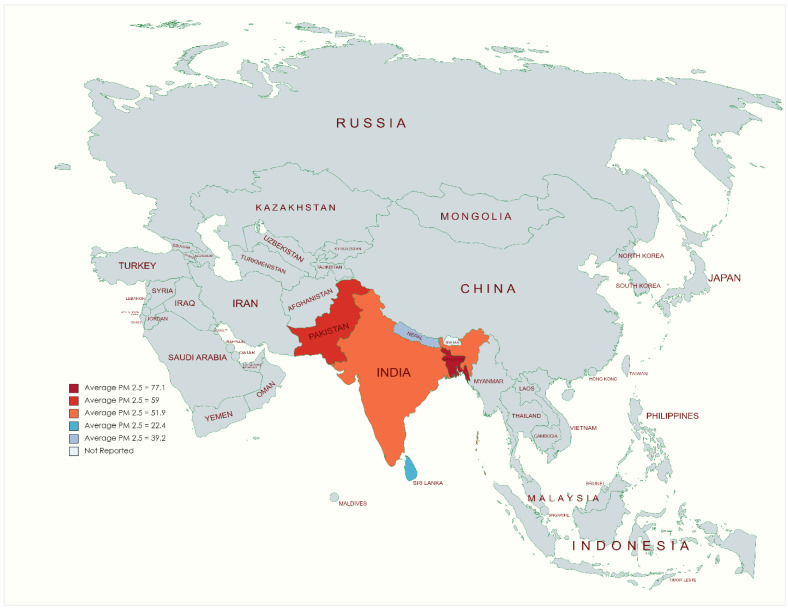
South Asian PM_2__.5_ (in µg/m^3^) yearly average visual representation.

## Data Availability

All data obtained for the purpose of this study are available online.

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
