# Peer review of "Air Quality, Pollution and Sustainability Trends in South Asia: A Population-Based Study"

_ijerph, 2022, doi:10.3390/ijerph19127534_

Round 1
Author Response
IJERPH-1750109: Air Quality, Pollution and Sustainability Trends in South Asia: A Population-Based Study
Reviewer 1:
Reviewer comment: This paper describes the air quality and the sustainability in South Asia, where most of the top-most polluted cities are located and the population is highly affected by air pollution. In order to perform the study, a population-based study was carried out. The sources of data are fully described and listed. After the discussion of the results, some recommendations are provided.
Response to reviewer comment: Thank you for your expert insight. Greatly appreciated.
Reviewer comment: The methods paragraph should be improved, step-by-step approach is not so clear.
Response to reviewer comment: The methods paragraph has been thoroughly edited and the step-by-step approach was drafted afresh for clarity purposes. Please review.
Reviewer comment:- Figure 2: The Pakistan graph has a different style (horizontal lines instead of vertical). It should be aligned with the others.
Response to reviewer comment: The graphs' design has been changed and they have been drafted afresh.
Reviewer comment: - Moreover, in this Figure, Bhutan, India, Nepal, Pakistan and Sri Lanka have exactly the same trends and the graphs don’t reflect what is written in the text. They have to be revised.
Response to reviewer comment: Thank you for noting out a grave error during drafting. Please review the updated figures. They reflect accurate trends for every country.
Reviewer comment: - The units of measure of y-axis are missing
Response to reviewer comment: The units of measure are listed as figure legend since inclusion in the figure directly would make them redundant and bulky: “i) carbon dioxide expressed in million tonnes. ii) total greenhouse gasses, iii) methane, and iv) nitrous oxide are all expressed in million tonnes of carbon dioxide-equivalents.” Please review.
Reviewer comment: Figure 4: Units of measure are missing. Not clear what annual change is. If it is a percentage, it should be explained better how it is calculated.
Response to reviewer comment: The chart represents Annual PM2.5 concentration in µg/m³, between 2018 and 2021. The annual change is discussed in the results, in terms of points change of annual PM2.5 concentrations before and during COVID-19. It is not a percentage, however, to avoid any confusion, the title and text has been updated.
Reviewer comment: Figure 5: - Units of measure (%) are missing.
Response to reviewer comment: The unit of measure is % and has been updated in the chart.
Reviewer comment: - Some data in the text don’t match the graph. For example, in the text it is written “The percentage of people using safely managed drinking water services was lowest in Nepal (17.6%), followed by Pakistan (35.8%) and Bhutan (58.5%)”, but Bhutan in the graph is lower than 40%.
Response to reviewer comment: All authors including myself have reviewed all the discrepancies in the data. They have been fully updated.
Reviewer comment: - Since it is a percentage, the x-axis limit should be 100, instead of 120.
Response to reviewer comment: The x-axis has been updated to 100.
Reviewer comment: - It is not clear in the text which year the data in the graph refers to.
Response to reviewer comment: The data is reported between the years 2015-2020 and every indicator has been updated with the corresponding year in the text. Please review.
Reviewer comment: In the whole text:- Chemical formulae must be written with subscripts. For example, CO2 has to be corrected with CO2 and NO2 with NO2
Response to reviewer comment: The chemical formulae have been corrected.
Reviewer 2 Report
This work estimates the emissions of GHGs and air pollutants in South Asia. While the topic fits the scope of IJERPH, there are some major issues that need to be addressed to further improve the quality of this manuscript.
1, Please use the subscripts and subscripts correctly.
2, The research objective is clearly described in Introduction, but the background and novelty are not well presented. The novelty of the work should be clearly addressed and discussed by comparing your study with existing research findings.
3, Following the question above, it is recommended to extend the analysis with a major number of references. It is important to classify and summarize these existing studies. This is helpful to identify the knowledge gap and highlight the novelty of this research.
4, Methods and Data are suggested to describe details.
5, Figs without any units of emissions or concentrations.
6. The main findings, contribution, and novelty should be clearly summarized
Author Response
IJERPH-1750109: Air Quality, Pollution and Sustainability Trends in South Asia: A Population-Based Study
Reviewer 2:
This work estimates the emissions of GHGs and air pollutants in South Asia. While the topic fits the scope of IJERPH, there are some major issues that need to be addressed to further improve the quality of this manuscript.
Reviewer comment: 1, Please use the subscripts and subscripts correctly.
Response to reviewer comment: They have now been placed correctly.
Reviewer comment: 2, The research objective is clearly described in Introduction, but the background and novelty are not well presented. The novelty of the work should be clearly addressed and discussed by comparing your study with existing research findings.
Response to reviewer comment: Thank you for your feedback. Searching through all current scholarly materials on the topic, the background has been expanded upon. Please review the first five paragraphs of the discussion. A comparison with ‘all’ existing literature has been added in addition to presenting the main findings.
Reviewer comment: 3, Following the question above, it is recommended to extend the analysis with a major number of references. It is important to classify and summarize these existing studies. This is helpful to identify the knowledge gap and highlight the novelty of this research.
Response to reviewer comment: Thank you for your comments. The goal of this paper is not to conduct a systematic review. This is a population based study. However, your comment is acknowledged in that we have updated the references.
Reviewer comment: 4, Methods and Data are suggested to describe details.
Response to reviewer comment: The methods and results have been crisped, please have a look.
Reviewer comment: 5, Figs without any units of emissions or concentrations.
Response to reviewer comment: All the figures have been clarified for units of emissions/x-y axis units. Please have a look.
Reviewer comment: 6. The main findings, contribution, and novelty should be clearly summarized
Response to reviewer comment: Please review the revised results and discussion. Clear summaries have been provided.